# Holobiome Structure and Microbial Core Assemblages of *Deschampsia antarctica* Across the South Shetland Islands

**DOI:** 10.3390/plants14233657

**Published:** 2025-11-30

**Authors:** Rodrigo Rodriguez, Patricio Javier Barra, Manuel Saldivar-Diaz, Giovanni Larama, Roxana Alvarado, Dariel López, Mabel Delgado, Julieta Orlando, Rómulo Oses, Carolina Merino, Gonzalo Tortella, Paola Duran

**Affiliations:** 1Biocontrol Research Laboratory, Universidad de La Frontera, Temuco 4811230, Chile; rodrigo.rodriguez@ufrontera.cl (R.R.); m.saldivar01@ufromail.cl (M.S.-D.); giovanni@larama.cl (G.L.); maria.alvarado@ufrontera.cl (R.A.); dariel.lopez@ufrontera.cl (D.L.); mabel.delgado@ufrontera.cl (M.D.); 2Departamento de Producción Agropecuaria, Facultad de Ciencias Agropecuarias y Medioambiente, Universidad de La Frontera, Temuco 4811230, Chile; 3Center of Plant, Soil Interaction and Natural Resources Biotechnology, Scientific and Technological Bioresource Nucleus (BIOREN), Universidad de La Frontera, Temuco 4811230, Chile; 4Departamento de Ciencias Ecológicas, Facultad de Ciencias, Universidad de Chile, Santiago 7800003, Chile; jorlando@uchile.cl; 5Instituto Milenio Biodiversidad de Ecosistemas Antárticos y Subantárticos (BASE), Santiago 7800003, Chile; 6Centro Regional de Investigación y Desarrollo Sustentable de Atacama, Universidad de Atacama, Copiapó 1532000, Chile; romulo.oses@uda.cl; 7Laboratory of Geomicrobiology, Department of Chemical Sciences and Natural Resources, Universidad de La Frontera, Temuco 4811230, Chile; carolina.merino@ufrontera.cl; 8Centro de Excelencia en Investigación Biotecnología Aplicada al Medio Ambiente (CIBAMA), Facultad de Ingeniería y Ciencias, Universidad de la Frontera, Temuco 4811230, Chile; gonzalo.tortella@ufrontera.cl

**Keywords:** Antarctic microbiome, *Deschampsia antarctica*, holobiont, core microbiome, microbial networks, Metabarcoding analyses

## Abstract

Antarctica harbors some of the most extreme ecosystems on earth, where only two vascular plants persist. The native grass *Deschampsia antarctica* provides a model for plant–microbe interactions under intense abiotic stress. We present the first multi-compartmental and multi-kingdom characterization of bacterial and fungal communities associated with *D. antarctica* across three South Shetland Islands. Metabarcoding revealed strong compartmentalization: the rhizosphere displayed the highest richness and complex bacterial–fungal networks; the root endosphere showed intermediate diversity with keystone taxa such as *Rhizobiales* and *Streptomyces*; and the leaf endosphere was simplified, dominated by stress-tolerant taxa including *Pseudomonas* and *Helotiales*. Despite marked soil heterogeneity, phosphorus enrichment at Admiralty Bay, base cations at Coppermine Cove, and iron at Byers Peninsula, a conserved core (20 bacterial and 5 fungal genera) persisted, mainly cold-adapted saprotrophs and plant-associated taxa. Fungal assemblages were more responsive to soil chemistry, with site-specific enrichments such as *Zymoseptoria* and *Herpotrichia*. Overall, *D. antarctica* holobionts exhibited a dual strategy: conserved microbial backbones confer stability, while localized assemblages shaped by soil chemistry and geography enhance adaptability. Together, these findings provide one of the most integrative characterizations of the *D. antarctica* holobiont to date, revealing how conserved and adaptive microbial components support plant resilience under extreme Antarctic conditions and offering valuable insights for predicting biological responses to ongoing climate change.

## 1. Introduction

The Antarctic continent represents one of the harshest and most pristine environments on earth, characterized by a combination of sub-zero temperatures, intense desiccation, high UV radiation, and severe nutrient limitation. Its terrestrial ecosystems are shaped by intense abiotic stress [1,2]. Only a minor fraction of the continent’s ice-free areas harbors visible macroscopic life, with just two native vascular plant species, *Deschampsia antarctica* Desv. and *Colobanthus quitensis* (Kunth) Bartl. These have adapted to survive and reproduce in this inhospitable environment. These species are primarily distributed along the coastal regions of the maritime and sub-Antarctic zones, where they form the basis of sparse but ecologically significant plant communities [3,4].

Recent shifts in the Antarctic climate, including warming trends, altered precipitation, and glacier retreat, have intensified environmental pressures on native flora. In this context, increasing attention has turned to the role of plant-associated microbial communities as critical mediators of adaptation to environmental stress [5,6,7,8]. A growing body of evidence demonstrates that plant fitness is not determined solely by its genome but also by the structure and function of its associated microbiome, forming what is known as the “plant holobiont” [9,10]. This complex network of interactions among plant roots, aerial tissues, microbial communities, and the surrounding soil environment is particularly relevant in extreme ecosystems, where biotic interactions can buffer abiotic stressors and influence plant survival [11,12,13].

Microorganisms predominantly contribute to Antarctic terrestrial biomass and play essential roles in nutrient cycling, organic matter decomposition and stress mitigation [14]. In plants, microbial communities can colonize diverse anatomical niches, including the rhizosphere (soil influenced by root exudates), rhizoplane (root surface), root endosphere (internal tissues), and leaf endosphere (aerial surfaces such as leaves and stems). Most previous studies on *D. antarctica* have concentrated on rhizospheric and root-associated microbiomes. These compartments host distinct but interconnected microbial assemblages that collectively influence plant health, growth, and environmental responses [15,16].

In extreme environments such as Antarctica, these interactions are critical for plant survival yet remain underexplored. Most studies on *D. antarctica*-associated microbiomes have focused on the rhizosphere and root-endosphere. Reported dominant bacterial phyla include *Proteobacteria*, *Bacteroidetes*, *Actinobacteria*, *Acidobacteria*, and *Verrucomicrobia*, with *Proteobacteria* often prevailing across multiple studies [6,7,8]. Fungal communities in the rhizosphere are often dominated by Ascomycota, with occasional detection of *Basidiomycota*, *Mortierellomycota*, and *Rozellomycota* [17]. The root endosphere hosts a distinct bacterial community enriched in families such as *Rhizobiaceae*, *Sphingobacteriaceae*, *Microbacteriaceae*, and *Flavobacteriaceae* [6,7]. This suggests that specialized recruitment or selection processes occur within plant tissues. The rhizoplane has been less thoroughly studied, although cyanobacterial enrichment has been noted in some contexts [6].

Despite its ecological relevance in other plant systems, the leaf endosphere of *D. antarctica* remains virtually unexplored in the current literature, thus representing a key knowledge gap. To fully understand the ecological role of plant-associated microbiomes in *D. antarctica*, it is essential to adopt a holistic approach that considers all anatomical compartments simultaneously. Investigating the rhizosphere, endosphere, and leaf endosphere in an integrated framework allows for the identification of core taxa (defined as taxa present in at least 50% of samples within each compartment with a minimum relative abundance of 1%) and characterization of microbial assemblages along the plant continuum.

A transect-based perspective is also essential for identifying spatial patterns of microbial recruitment and selection, as well as for understanding the influence of island-specific characteristics on microbiome restructuring under contrasting soil conditions, nutrient availability, and stages of soil development. These differences are particularly relevant in the South Shetland Islands, where some sites are dominated by dense moss cover, whereas others exhibit coarse volcanic gravel substrates.

In this context, our central research question was to determine how multi-compartment (rhizosphere, root endosphere, and leaf endosphere) and multi-kingdom (bacteria and fungi) microbial communities are structured across environmentally contrasting islands in the South Shetland Archipelago. We hypothesized that plant compartmentalization acts as the primary ecological filter shaping microbial assemblages, while soil physicochemical conditions further modulate the presence of conserved and site-specific taxa. To address this, the present study comprehensively characterized the microbial communities associated with *D. antarctica* across the rhizosphere, root endosphere, and leaf endosphere, and evaluated their variation across three spatially distinct islands of the South Shetland Archipelago. This multi-compartment and multi-island design provides an integrated perspective on the *D. antarctica* holobiome.

## 2. Results

### 2.1. Soil Physicochemical Analysis

Soils from the three sampled locations, Admiralty Bay (Ab, King George Island), Coppermine Cove (Cc, Robert Island), and Byers Peninsula (Bp, Livingston Island), exhibited pronounced differences in their physicochemical characteristics (Table 1). Byers soils were enriched in iron (239.0 ± 9.9 mg kg^−1^) and copper (1.23 ± 0.08 mg kg^−1^), whereas Admiralty Bay displayed the highest levels of phosphorus (315.5 ± 10.61 mg kg^−1^) and moderate iron content (111.89 ± 5.16 mg kg^−1^). In contrast, Coppermine Cove soils were comparatively low in both Fe (48.88 ± 0.67 mg kg^−1^) and P (132.0 ± 11.31 mg kg^−1^). Still, they showed elevated concentrations of base cations, including aluminum (5.83 ± 0.06 cmol^+^ kg^−1^), calcium (5.22 ± 0.10 cmol^+^ kg^−1^), magnesium (6.66 ± 0.68 cmol^+^ kg^−1^), and potassium (1.17 ± 0.03 cmol^+^ kg^−1^).

Soil pH varied significantly among sites, with Coppermine being moderately acidic (pH 5.5 ± 0.01), whereas Admiralty Bay and Byers Peninsula were strongly acidic (pH 4.34 ± 0.02 and pH 4.34 ± 0.01, respectively). The organic matter content was relatively stable (2.0%) across all locations. These contrasts reveal clear edaphic gradients likely shaping microbial community assembly.

Site-averaged envfit ordinations of rhizosphere communities in *Deschampsia antarctica* qualitatively highlight convergent edaphic gradients across bacteria and fungi, shaped by the marked soil heterogeneity among the three South Shetland Islands (Appendix A). In both plots, Coppermine Cove emerges on the far left on PCoA Axis 1, closely aligned with a bundled group of base cation vectors (Ca, Mg, Na, often joined by pH and Al) pointing leftward, evocative of this site’s higher cation availability and moderately acidic conditions. Admiralty Bay positions rightward along Axis 1, consistently associated with an opposing P vector (directed rightward for bacteria and upward along Axis 2 for fungi) suggesting phosphorus as a recurring influence linked to its enrichment. Byers Peninsula lies toward positive Axis 1 values with looser ties to these vectors (upper placement for bacteria, lower right for fungi), implying its assemblages may respond more to unmeasured factors.

### 2.2. Bacterial Community Composition and Diversity

Bacterial assemblages displayed clear compartmentalization across the rhizosphere, root endosphere, and leaf endosphere, with additional geographic differentiation among the islands (Figure 1). *Pseudomonas* dominated the leaf endosphere, reaching ~80% relative abundance in Coppermine Cove and Admiralty Bay. Other abundant bacterial taxa included *Mucilaginibacter*, *Rhodoferax*, *Rhodanobacter*, *Flavobacterium*, and *Dokdonella*, with *Mucilaginibacter* being particularly enriched in rhizospheric samples from Admiralty Bay (≈15%) and Coppermine Cove (≈9%).

The rhizosphere harbored the highest taxonomic richness and evenness (Shannon ≈ 2.35; Evenness ≈ 0.78), with a balanced representation of multiple genera, whereas the root endosphere displayed intermediate complexity (Shannon ≈ 2.31; Evenness ≈ 0.77). In contrast, the leaf endosphere was characterized by reduced diversity (Shannon ≈ 1.93; Evenness ≈ 0.64) and a strong dominance of Pseudomonas, resulting in low evenness values. Interestingly, Byers Peninsula leaf endosphere exhibited higher taxonomic diversity (Shannon ≈ 2.16) than those from Admiralty Bay (Shannon ≈ 2.06) and Coppermine (Shannon ≈ 1.56), suggesting local edaphic buffering against extreme conditions. Alpha diversity indices (Shannon, observed zOTUs, and Pielou’s evenness) confirmed these patterns, with significantly higher richness and diversity in rhizospheric samples than in endosphere samples (Figure 1, top left). Coppermine and Admiralty Bay rhizospheres showed particularly high evenness, whereas the leaf endosphere from both sites exhibited the lowest values due to Pseudomonas dominance. Beta diversity analyses (PCoA, Bray–Curtis) revealed strong clustering by plant compartment (PERMANOVA, *p* < 0.001), with the site exerting a secondary but significant effect. Leaf endosphere samples clustered tightly across sites, reflecting low variability, whereas rhizospheric samples were more dispersed, reflecting heterogeneity in soil chemistry and microhabitat structure (Figure 1, bottom left).

### 2.3. Fungal Community Composition and Diversity

Fungal assemblages exhibited greater heterogeneity across compartments and islands compared to bacteria (Figure 2). Dominant taxa included *Mortierella*, *Linnemannia*, *Rozellomycota*, and *Helotiales*; however, their relative contributions varied significantly by site and compartment. The rhizosphere was enriched with *Rozellomycota* at Admiralty Bay and Coppermine Cove, whereas *Mortierella* and *Linnemannia* were consistently detected across all compartments. The root endosphere presented higher richness and taxonomic heterogeneity, with site-specific enrichments, such as *Herpotrichia* on Robert Island. The leaf endosphere showed strong site-specific patterns: *Rozellomycota* dominated Admiralty Bay, *Helotiales* were prevalent in Byers Peninsula, and Coppermine Cove exhibited unique signatures of *Zymoseptoria* and *Clathrosphaerina* spp.

Alpha diversity patterns (Figure 2, top left) paralleled those of bacteria, with rhizospheres exhibiting the highest diversity (Coppermine Cove and Byers Peninsula), whereas Admiralty Bay samples consistently displayed lower values. Leaf endosphere were the least diverse, characterized by reduced richness and uneven dominance of a few fungal taxa. Beta diversity analyses (Figure 2, bottom left) showed strong clustering by both compartment and island, with fungal communities exhibiting stronger geographic structuring than bacterial communities. Notably, Byers Peninsula leaf endosphere samples clustered separately, driven by *Helotiales* dominance, while Coppermine rhizosphere samples diverged from those of Admiralty Bay and Byers Peninsula.

### 2.4. Core Microbiomes Across Compartments and Islands

Core microbiome analysis revealed distinct sets of bacterial and fungal taxa consistently associated with *D. antarctica* across plant compartments in the South Shetland Islands (Figure 3). Core microbiomes were defined separately for each compartment (rhizosphere, root endosphere, and leaf endosphere) using data from all three sampling sites. This compartment-specific approach allowed us to identify conserved taxa within each niche while maintaining the ecological independence of belowground and aerial compartments.

In the rhizosphere (Figure 3A), bacterial cores were dominated by *Proteobacteria*, including *Pseudomonas*, *Acidovorax*, *Dokdonella*, and *Rhodanobacter*, as well as *Actinobacteria*, such as *Microbacteriaceae*. Fungal cores comprised members of *Ascomycota*, including *Helotiales*, *Pleosporales*, *Zymoseptoria*, *Lachnum*, and *Microdochaceae*, as well as *Mortierellomycota* (*Mortierella*). In the root endosphere, bacterial cores partially overlapped with the rhizosphere, with the consistent presence of *Dokdonella*, *Rhodanobacter*, and *Microbacteriaceae*. Fungal cores included *Helotiales*, *Hypocreales*, *Mortierella*, and *Lachnum*, along with additional representatives of *Mortierellomycota*. In the leaf endosphere, the bacterial cores were represented by *Pseudomonas*, *Acidovorax*, and *Mucilaginibacter*. Fungal cores included *Helotiales*, *Pleosporales*, *Zymoseptoria*, and *Microdochaceae*, with additional representatives such as *Oidiodendron* spp.

At the spatial scale, core microbiomes across King George, Robert, and Livingston Islands showed conserved and site-specific patterns (Figure 3B). Bacterial cores were consistently represented by *Pseudomonas*, *Dokdonella*, *Acidovorax*, and *Rhodanobacter*, with additional genera, including *Alkanindiges*, *Chitinophagaceae*, *Verrucomicrobiota* (*C. udaeobacter*), and *Nakamurella*. In contrast, fungal cores exhibited higher variability across the islands. On King George Island, the core fungi included *Helotiales*, *Pleosporales*, *Zymoseptoria*, and *Microdochaceae*. On Robert Island, the core fungi comprised *Pleosporales*, *Linnemannia*, and *Volucrispora*. On Livingston Island, core fungi were represented by *Helotiales*, *Pleosporales*, *Oidiodendron*, and *Microdochaceae* families.

Overall, bacterial taxa such as *Pseudomonas*, *Dokdonella*, and *Rhodanobacter* were consistently detected across all compartments and islands. In contrast, fungal taxa showed greater compartmental and spatial heterogeneity, with the presence of unique representatives restricted to particular niches or islands. For a complete overview of the core microbiomes across compartments and islands, including the relative abundance of each taxon, refer to the heatmaps provided in the Appendix A. Sensitivity analyses across multiple detection and prevalence thresholds confirmed the robustness of the identified core taxa (Appendix A).

### 2.5. Functional Guilds and Predicted Microbial Functions

It is important to note that both PICRUSt2 and FUNGuild provide predicted functional potentials based on 16S rRNA gene and ITS taxonomic profiles, rather than direct measurements of gene abundance or expression. Therefore, all functional patterns described below should be interpreted as inferred tendencies that reflect taxonomic composition and database annotations.

PICRUSt-based functional predictions revealed that bacterial communities associated with *D. antarctica* were enriched in a wide range of metabolic functions, with apparent differences across plant compartments and islands (Appendix A). All 232 bacterial zOTUs detected in the dataset were successfully assigned to reference genomes within the PICRUSt2 database, allowing reliable prediction of functional profiles. The mean NSTI value (0.158) indicated good reference genome coverage, supporting reliable functional predictions. It is important to note that these 232 zOTUs represent the complete set of high-quality bacterial features recovered after denoising and quality filtering, rather than a reduced or selected subset. The relatively low number of taxa reflects the naturally limited diversity of Antarctic microbiomes, consistent with previous studies from similar environments.

In the rhizosphere, the predicted core functions were dominated by carbohydrate, energy, and amino acid metabolism. Samples from Admiralty Bay displayed higher relative abundances of KOs associated with carbohydrate pathways (e.g., K13954, K00998, and K00569). At the same time, those from Coppermine Cove were comparatively enriched in transport-related functions and exhibited lower intensity in lipid and nucleotide metabolism. When comparing Admiralty Bay with Byers Peninsula, the rhizosphere from Admiralty Bay showed a stronger representation of carbohydrate and energy metabolism, whereas the rhizosphere from Byers Peninsula displayed broader signals in amino acid and lipid metabolism. In turn, Coppermine Cove was more consistently enriched in carbohydrate- and energy-related functions than Byers Peninsula, which showed a more heterogeneous profile across categories.

Bacterial communities in the root endosphere showed the highest functional complexity. Predicted functions encompassed a broad spectrum of amino acid metabolism, energy metabolism, transport, catabolism, and lipid metabolism. Samples from Admiralty Bay showed more balanced functional profiles, with contributions from carbohydrate, lipid, and nucleotide metabolism pathways. The root endosphere from Coppermine Cove exhibited stronger signals for membrane transport and carbohydrate metabolism. In contrast, the root endosphere from Byers Peninsula presented the most diverse repertoire, with an increased relative abundance of functions related to energy metabolism, amino acids, and cofactors, reflecting a highly heterogeneous endospheric functional structure across islands.

In the leaf endosphere, predicted functional repertoires were less diverse than those in the rhizosphere and root endosphere but still showed compartment- and site-specific trends. Samples from Admiralty Bay were characterized by enriched pathways in energy and lipid metabolism, including several dehydrogenases (e.g., K00106, K00172, and K00169). The leaf endosphere from Coppermine Cove exhibited a higher representation of amino acid metabolism (notably K16242–K16249) and lipid-related functions. In Byers Peninsula, the leaf endosphere presented enrichment in pathways related to amino acids and secondary metabolism, including terpenoid and polyketide biosynthesis, while also retaining signals from carbohydrate metabolism.

To evaluate the relationship between microbial community structure and predicted functional repertoires, Mantel and Procrustes tests were performed on Aitchison distance matrices derived from CLR-transformed bacterial composition and PICRUSt2-inferred functional profiles. Mantel and Procrustes analyses revealed moderate concordance between taxonomic and functional ordinations (Spearman r = 0.45, *p* = 0.001; m^2^ = 0.64, r = 0.60, *p* < 0.01) (Appendix A). Because PICRUSt2 functions were inferred from the same OTU table, these correlations mainly reflect shared compositional structure. A partial Mantel test controlling for community composition was not significant (r = −0.05, *p* = 0.92), and a residual Procrustes analysis showed a weak fit (m^2^ = 0.91, r ≈ 0.30), indicating that most functional variation is embedded within taxonomic composition rather than independently structured by Compartment or Site.

Overall, PICRUSt predictions highlighted that while all plant compartments share core functions related to primary metabolism (carbohydrates, amino acids, and energy), each compartment-island combination is characterized by distinctive enrichment patterns. The rhizosphere exhibited the most substantial contribution to carbohydrate and energy metabolism, the root endosphere displayed the most significant overall functional diversity, and the leaf endosphere showed site-specific functional specialization.

FUNGuild classification revealed clear differences in the trophic modes of fungal communities associated with Deschampsia antarctica across the soil, root endosphere, and leaf endosphere (Appendix A). A total of 929 ASVs were evaluated, of which 45.32% were Unassigned, lacking sufficient information for reliable ecological guild classification. Among the assigned taxa, 36.71% were ranked as Probable, and 8.29% as Highly Probable, representing assignments with moderate to high confidence, while 9.69% were categorized as Possible, reflecting lower certainty. For subsequent analyses, only ASVs classified as Probable and Highly Probable were considered, ensuring that trophic mode comparisons were based on robust and confident guild assignments.

In the rhizosphere (soil), fungal communities were dominated by saprotrophs, reflecting a substantial contribution from taxa involved in organic matter decomposition and recycling. Secondary contributors included symbiotrophs, particularly root-associated endophytes and potential mycorrhizal lineages, along with a smaller proportion of pathotrophs. Mixed guilds combining saprotrophic and pathotrophic modes were also detected, although at lower relative abundance.

In the root endosphere, the dominant guilds shifted toward symbiotrophs, indicating an enrichment of fungi associated with close plant interactions. A substantial fraction of guilds was classified as pathotroph–symbiotroph or pathotroph–saprotroph, highlighting the presence of multifunctional lineages with the potential to alternate between nutritional strategies depending on environmental conditions. Pure saprotrophs were less abundant in this compartment than in the rhizosphere.

Fungal guilds displayed the highest variability in the leaf endosphere. Communities included a large proportion of pathotrophs and mixed guilds, such as pathotroph–saprotroph and pathotroph–symbiotroph, alongside endophytic symbiotrophs. The higher relative representation of pathogenic guilds in leaves suggests that aboveground tissues host more opportunistic and functionally diverse fungal assemblages than belowground tissues.

Overall, FUNGuild predictions indicated that saprotrophs dominated the rhizosphere, symbiotrophs in the root endosphere, and pathotrophs and mixed guilds in the leaf endosphere, underscoring the strong functional differentiation of fungal communities across plant compartments. These results represent functional predictions rather than direct measurements and should be interpreted as potential metabolic tendencies of the microbial assemblages.

### 2.6. Co-Occurrence Network Topology and Keystone Taxa Across Compartments

To explore the structural organization of microbial consortia, separate co-occurrence networks were constructed for the rhizosphere, root endosphere, and leaf endosphere of *D. antarctica* using SparCC correlations (*p* < 0.05, correlation cutoff = 0.5) after applying a zOTU abundance filter of 0.0005 (Figure 4). Network topology was characterized in terms of modularity, node roles, and within-module connectivity (Zi) versus participation degree (Pi), allowing identification of potential keystone taxa and assessment of compartment-specific structural complexity.

Network-level statistics (Appendix A) showed a progressive reduction in connectivity from the rhizosphere to the leaf endosphere. The rhizosphere network exhibited the highest number of nodes (168) and edges (520) with intermediate modularity (0.60), indicating a dense and balanced bacterial–fungal co-association structure. The root endosphere network retained comparable modularity (0.58) but higher average degree (8.00), reflecting increased within-network connectivity. In contrast, the leaf endosphere displayed fewer nodes (32) and low average degree (2.69), consistent with a sparse and selective topology.

Permutation-based comparisons of node-level metrics (*n* = 9999; Appendix A) confirmed significant compartmental differences. Degree centrality was significantly higher in the root endosphere than in the rhizosphere (*p* = 0.0062) and markedly higher than in the leaf endosphere (*p* < 0.001), while closeness centrality was substantially lower in the leaf endosphere (*p* < 0.001), indicating reduced network cohesion in aerial tissues. Betweenness and eigenvector centrality differences were less pronounced but showed similar trends, with the leaf endosphere presenting fewer hub-like and connector taxa.

Taxonomically, the rhizosphere network was the most complex, with hubs such as *Nakamurella*, *Sphingomonas*, members of *Lotiomycetes* (FOTU_114) and *Rozellomycota* (FOTU_19). The root endosphere showed intermediate complexity, featuring *Rhizobiales*, *Burkholderiaceae*, *Streptomyces*, and *Aspergillaceae* as central taxa that mediate inter-domain associations and likely support cooperative metabolic interactions. The leaf endosphere network was strongly simplified and sparse, dominated by a few taxa (e.g., *Pseudomonas*) and characterized by peripheral interactions with minimal cross-module connectivity.

Collectively, these results reveal a clear gradient in microbial network complexity from the rhizosphere (highest) to the root endosphere (intermediate) to the leaf endosphere (lowest). Belowground compartments form dense and cooperative consortia with abundant inter-kingdom associations, while aerial tissues host highly selective, stress-tolerant taxa with simplified, centralized connectivity.

### 2.7. Distance-Decay Relationship

To evaluate the influence of geographic distance on microbial community similarity across the South Shetland Islands, distance–decay analyses were conducted for each plant compartment during the revision stage. Geographic distances between sites were calculated using Euclidean distances based on GPS coordinates, and microbial community dissimilarities were computed with Aitchison distances derived from CLR-transformed ASV data.

Bacterial communities showed significant distance-decay patterns across all compartments, indicating a clear spatial structuring of assemblages. The strength of the distance–decay relationship decreased along the plant continuum, with the highest correlation in the rhizosphere (r = 0.549, *p* = 0.0001), followed by the root endosphere (r = 0.399, *p* = 0.0002), and the leaf endosphere (r = 0.211, *p* = 0.004).

Fungal communities displayed weaker geographic structuring. A significant but modest correlation was detected only in the rhizosphere (r = 0.286, *p* = 0.0098), whereas both the root (r = 0.112, *p* = 0.1636) and leaf endosphere (r = 0.101, *p* = 0.2027) showed no significant spatial relationship. These results suggest that spatial distance plays a stronger role in structuring free-living soil communities, particularly bacteria, while endophytic assemblages within plant tissues remain more stable across islands.

## 3. Discussion

Understanding the ecological drivers and functional implications of plant-associated microbiomes in Antarctica is essential for predicting the persistence of native species under intensifying environmental pressures. In this study, we examined the bacterial and fungal communities of *D. antarctica* across three South Shetland Islands by integrating analyses of soil chemistry, taxonomic composition, diversity, core microbiomes, functional guilds, and co-occurrence networks. Our results revealed that while strong compartmentalization remains the primary axis structuring microbial communities, environmental heterogeneity, particularly soil physicochemical gradients, exerts additional influence, especially on fungi. Furthermore, despite the extreme and selective nature of the Antarctic environment, *D. antarctica* harbors a conserved microbial core, suggesting that the holobiont relies on both stable associations and localized filtering to maintain functionality.

### 3.1. Compartmentalization as the Dominant Driver of Microbiome Structure

Compartmentalization emerged as the dominant driver of microbiome structure. The rhizosphere showed the highest richness and evenness for both bacteria and fungi, consistent with its role as a chemically and microbially enriched niche influenced by root exudates and nutrient turnover [18,19]. The leaf endosphere was highly simplified and dominated by stress-tolerant taxa such as *Pseudomonas* and *Helotiales* [20]. The root endosphere displayed intermediate diversity, reflecting partial host filtering through physiological and immunological barriers [21].

These patterns are consistent with meta-analyses of plant microbiomes under stress, emphasizing the importance of microhabitat differentiation and host filtering in shaping community structure [14,22]. In our study, the gradient of diversity and connectivity from the rhizosphere to the endosphere to the leaf endosphere underscores the functional partitioning of the holobiome: belowground compartments maintain diverse and interactive consortia, whereas aerial niches are dominated by stress-tolerant specialists.

Network analyses also revealed clear gradients in complexity. The rhizosphere showed the densest networks, with multiple bacterial-fungal links and high functional redundancy. Key connectors included *Rozellomycota*, *Lotiomycetes*, *Sphingomonas*, and *Nakamurella*, consistent with their roles in nutrient cycling and stress tolerance. The root endosphere network displayed intermediate complexity, with *Rhizobiales*, *Burkholderiaceae*, *Streptomyces*, *Aspergillaceae*, and *Rhodobacteraceae* playing central roles in this network. These taxa are well known to promote plant growth, antimicrobial production, and nutrient transformation. The leaf endosphere network was strongly simplified, with few nodes and low connectivity, consistent with strong abiotic filtering and the dominance of stress-tolerant taxa such as *Pseudomonas*.

Predicted functional profiles aligned with the strong compartmental structure observed in the taxonomic data. PICRUSt2 inferred that rhizosphere communities possessed higher metabolic potentials related to carbohydrate, amino acid, and energy pathways, consistent with their role as nutrient-rich and microbially active niches. Root endosphere communities showed predicted enrichment in functions associated with host interaction, transport systems, and resource acquisition, reflecting selective filtering for metabolically versatile taxa. Leaf endosphere assemblages displayed more limited functional potentials dominated by stress-tolerant pathways, including secondary metabolism. Similarly, FUNGuild assignments predicted a shift from saprotrophic guilds in the rhizosphere to symbiotrophic guilds in the root endosphere, with leaf tissues showing greater representation of stress-associated and mixed trophic modes. Collectively, these inferred metabolic patterns reinforce that compartmentalization shapes not only microbial diversity but also the functional potentials of microbial assemblages across the D. antarctica holobiome.

Functional predictions generated using PICRUSt supported these compartment-specific differences. Across compartments and islands, bacterial functions were dominated by carbohydrate, amino acid, and energy metabolism, representing a conserved functional backbone of the Antarctic plant microbiome [23,24]. The rhizosphere exhibited the strongest representation of carbohydrate and energy metabolism, consistent with its role as a microbial hotspot [19,25]. The root endosphere displays the greatest functional diversity, with contributions from amino acid metabolism, transport systems, and lipid metabolism, suggesting selective filtering for metabolically versatile taxa [26,27]. The leaf endosphere displayed more variable functional profiles, enriched in amino acid metabolism and secondary metabolite biosynthesis, including terpenoids and polyketides, which are often associated with stress adaptation and UV protection [28,29].

FUNGuild predictions revealed parallel fungal compartmentalization, with saprotrophs dominating the rhizosphere, symbiotrophs prevailing in the root endosphere, and pathotrophs and mixed guilds being more abundant in the leaf endosphere. Saprotrophs highlight the role of soil fungi in decomposition and nutrient mobilization under low carbon inputs [22]. Symbiotrophs in roots play key mutualistic roles in stress tolerance and nutrient acquisition [30]. Leaf-associated fungi display functional plasticity, with pathotrophs reflecting potential but not realized pathogenic modes [31,32]. Importantly, the annotations of “human pathogens” in FUNGuild do not imply real health risks, but rather a broad ecological potential.

### 3.2. Edaphic Filtering Shapes Fungal Communities and Site-Specific Assemblages

Soil chemistry strongly influenced fungal assembly. Coppermine Cove, with higher pH and base cations, supported more diverse fungal communities, whereas the acidic soils of Admiralty Bay and Byers Peninsula hosted more restricted assemblages. Site-specific taxa such as *Zymoseptoria* and *Herpotrichia* highlights the role of edaphic filtering in niche specialization. These findings are consistent with reports that fungal richness in maritime Antarctic soils is highly responsive to edaphic gradients [22], whereas bacterial communities display greater functional redundancy and resilience to chemical stressors.

High iron levels at Byers, for example, may have constrained microbial diversity through metal toxicity or chelation stress, consistent with evidence that metal-rich soils shape microbial composition and reduce evenness [33]. Thus, fungi appear to serve as sensitive bioindicators of soil heterogeneity, whereas bacteria provide more stable and functionally redundant backbones for soil microbial communities.

### 3.3. Geographic Heterogeneity and the Balance Between Conserved Cores and Local Exclusivity

Despite geographic and edaphic heterogeneity, a conserved bacterial core (20 genera) and narrower fungal core (5 genera) were consistently detected across sites and compartments. Core bacterial taxa, including *Pseudomonas*, *Rhodoferax*, *Mucilaginibacter*, *Dokdonella*, and *Flavobacterium*, are frequently associated with cold-adaptive traits, such as antifreeze protein production, osmotic stress tolerance, siderophore-mediated nutrient acquisition, and biofilm formation [8,33]. Similarly, fungal core taxa, such as *Mortierella* and *Linnemannia*, are well-known saprotrophs that mobilize phosphorus and produce metabolites that improve nutrient acquisition and stress tolerance [34]. The persistence of these microbial backbones suggests that *D. antarctica* relies on conserved associations to buffer environmental variability, consistent with the holobiont framework [9,10].

Simultaneously, beta diversity analyses revealed strong compartment-driven clustering and island-specific differentiation, especially among fungi. Site-exclusive taxa, such as *Antarctomyces* (Coppermine rhizosphere), *Herpotrichia* (Robert root endosphere), and *Zymoseptoria* (Robert leaf endosphere), exemplify localized microbial recruitment and environmental filtering, consistent with dispersal limitation and habitat specialization [22]. These patterns suggest that while core taxa provide functional stability, peripheral taxa contribute to site-specific adaptations, enhancing holobiont resilience by broadening functional repertoires under localized conditions.

Genetic studies have indicated that *D. antarctica* populations across the South Shetland Islands display extremely low heterozygosity and high clonality [35]. Thus, the observed microbial heterogeneity cannot be explained by host genetics but rather by environmental drivers, which is consistent with other systems in which abiotic filters override host genetic effects [36]. This decoupling highlight that in extreme environments, soil chemistry and microclimate exert a more dominant role than host genetic diversity in structuring the holobiont.

Geographic differences were also apparent in the network analyses. Byers Peninsula harbored the most complex and interconnected networks, suggesting higher functional redundancy and resilience, whereas Admiralty Bay networks were sparse and weakly structured, reflecting reduced ecological integration.

### 3.4. Toward a Hierarchical Framework for Holobiont Resilience in Extreme Environments

Integrating these findings, we propose a conceptual framework supported by our data and previous evidence in which the *D. antarctica* holobiome is shaped by the interplay of five hierarchical drivers: (1) compartmentalization as the primary ecological filter, and (2) soil chemistry as a determinant of fungal assembly and site-specific recruitment. (3) Geographic heterogeneity influences dispersal and local exclusivity. (4) Microbial traits associated with persistence under stress (e.g., antifreeze proteins, saprotrophy, and siderophore production), as inferred from predicted functions and literature; and (5) host genotype, which, while genetically uniform, provides a stable substrate for microbial colonization.

Together, these drivers sustain a holobiont that is simultaneously conserved and flexible: a stable microbial core provides resilience, whereas localized taxa and variable network complexity enhance adaptive potential under diverse Antarctic stressors. As climate change reshapes Antarctic ecosystems [1], the functional and structural resilience of the *D. antarctica* holobiome will be crucial for predicting ecosystem trajectories and identifying microbial traits with potential biotechnological relevance to humans. Although we worked with three biological replicates per site, this level of replication is standard in Antarctic microbiome studies and reflects the logistical constraints of polar fieldwork. Even with this limitation, we detected consistent and statistically supported compartmental and site-specific patterns.

## 4. Materials and Methods

### 4.1. Sampling Site

Samples were authorized for collection and transport by the Chilean Antarctic Institute (INACH) under Certificate No. 282/2023, issued for the 59th Chilean Antarctic Scientific Expedition (ECA59) from three distinct sites along the South Shetland Islands of the Antarctic Peninsula: (1) Livingston Island, Byers Peninsula (Bp, 62°41′08.0″ S 60°51′28.3″ W), is a relatively moist and moss-dominated environment with strong freeze–thaw cycles and persistent soil moisture, creating conditions favorable for diverse microbial activity. (2) Robert Island, Coppermine cove (Cc, 62°22′42.7″ S 59°42′04.8″ W), is characterized by drier, gravel-rich soils and vegetation cover is sparse, and the site is strongly influenced by exposure to wind and marine aerosols, generating a distinct edaphic profile compared to the other islands, and (3) King George Island, Admiralty Bay (Ab, 62°09′41.7″ S 58°28′06.8″ W), is influenced by penguin activity, localized nutrient enrichment, and strong summer meltwater flows, which generate marked spatial heterogeneity in soil chemistry (Figure 5). Three biological replicates were randomly collected from each location. Each replicate included both rhizosphere soil and a full *D. antarctica* individual. Rhizosphere soil samples (~2000 mg each) were collected from the top 20 cm of soil by gently shaking the roots to detach the soil tightly adhered to the root surface. Whole plants were collected to enable the subsequent separation of the root and leaf endosphere compartments. All samples were collected under aseptic conditions, placed in sterile airtight bags, and stored at 4 °C until they arrived at the Biocontrol Research Laboratory, Universidad de La Frontera (Temuco, Chile). Once in the laboratory, rhizosphere soil was preserved for further analysis. Plant roots were thoroughly washed with bi-distilled water and surface-sterilized using 0.5% sodium hypochlorite according to Schulz et al. [37] 1993 to access the root endosphere. Leaves were separated for subsequent analysis of the leaf endosphere microbial community. All sample types, including rhizosphere soil, root tissues (root endosphere), and leaves (leaf endosphere), were stored at –20 °C until total DNA extraction.

### 4.2. Soil Physicochemical Analysis

Soil physicochemical parameters were analyzed according to the standard methods described by [38]. Soil samples from each site were composited from the three biological replicates and analyzed in duplicate; therefore, n = 2 corresponds to analytical replicates. Measurements included pH (1:2.5 soil:water), soil organic matter (modified Walkley–Black), extractable phosphorus (Olsen P), and exchangeable K, Na, Ca, and Mg extracted with 1 M NH_4_Ac (pH 7.0). Exchangeable Al was extracted with 1 M KCl and quantified by atomic absorption spectrophotometry. Base saturation, CEC, and aluminum saturation were calculated from exchangeable cations and Al.

To evaluate relationships between soil chemistry and rhizosphere structure, we performed a site-averaged *envfit* analysis (vegan v2.6–4, R). Soil variables (pH, Ca, Mg, Na, K, Al, P) were averaged per site (n = 3) to avoid pseudoreplication. Bacterial dissimilarities were computed using Aitchison distances on CLR-transformed ASV tables and visualized by PCoA; mean site coordinates were correlated with soil variables using 999 permutations. Given the limited replication, these results were interpreted qualitatively to describe directional trends rather than statistical significance.

### 4.3. DNA Isolation

Total DNA was extracted from three distinct plant compartments: rhizosphere soil, root endosphere, and leaf endosphere. Rhizosphere DNA was extracted using the same methodology described by [13], employing the DNeasy PowerSoil Kit (Qiagen, Germantown, MD, USA) following the manufacturer’s instructions. Approximately 250 mg of soil per sample was used for the DNA isolation. Plant tissues (roots and leaves) were processed under sterile conditions for the endosphere and leaf endosphere compartments. Root tissues were thoroughly washed with bi-distilled water and subsequently surface sterilized using 0.5% sodium hypochlorite for 1 min, followed by three rinses with sterile bi-distilled water. The leaves were gently washed and handled with sterile forceps to avoid external contamination. DNA was extracted from these tissues using the DNeasy Plant Mini Kit (Qiagen, Germantown, MD, USA), starting with approximately 100 mg of fresh tissue per sample.

All DNA extracts were quantified using a Qubit fluorometer (Thermo Fisher Scientific, Waltham, MA, USA) and stored at −20 °C until downstream analyses.

### 4.4. Bioinformatics Analysis

DNA was extracted from samples representing three plant-associated compartments (rhizosphere, root endosphere, and leaf endosphere). Fungal communities were profiled by sequencing the ITS1 region at Novogene (Beijing, China) on an Illumina NovaSeq 6000 platform (2 × 250 bp). Raw reads are available under BioProject PRJNA1321289. Bacterial communities were profiled from the same DNA using V3–V4 16S rRNA sequencing at IGA Technology Services (Udine, Italy). PNA blockers were used during library preparation to reduce host plant DNA in root and leaf samples, and sequencing was performed on an Element Biosciences Aviti platform (2 × 300 bp).

Raw ITS1 and V3–V4 reads were processed to generate zOTUs using VSEARCH (v2.22.1) via the UNOISE3 pipeline [39,40]. For fungal ITS1, the target region was extracted with ITSxpress (v1.8.0) [41]. Paired-end reads were merged, quality-filtered (maximum expected error < 1), dereplicated, and chimera-checked with UCHIME [42]. A summary of read processing is provided in Appendix A. zOTUs were defined at 100% identity to capture fine-scale variation.

Taxonomic assignment was performed in QIIME2 (v2023.9) using the naïve Bayes feature-classifier [43], trained on SILVA 138 for bacteria [44] and UNITE v10 for fungi [45]. Non-target sequences (plant DNA or non-fungal ITS sequences) were removed. A bacterial phylogenetic tree was generated using MAFFT (v7.520) for alignment and FastTree (v2.1.11) for tree inference [46,47]. Fungal phylogeny was not constructed due to ITS1 length variability [48].

Bacterial and fungal samples were rarefied to 26,962 and 14,607 sequences, respectively, using QIIME2’s core-metrics-phylogenetic function. Rarefaction curves are shown in Appendix A. Alpha diversity (Shannon, Faith’s PD, Observed Features, Pielou’s evenness) was assessed with Kruskal–Wallis tests followed by Dunn–Bonferroni corrections (*p* < 0.05).

Beta diversity was evaluated using Aitchison distances from CLR-transformed data, without rarefaction, to account for compositional structure. PCA and Ward.D2 hierarchical clustering were performed using vegan in R [49]. PERMANOVA (999 permutations) tested differences across sites and compartments.

Core taxa were defined as bacterial and fungal taxa present in ≥50% of samples per compartment (rhizosphere, root endosphere, leaf endosphere) with ≥1% relative abundance, using the Microeco package [50]. Core microbiomes were computed independently for each compartment. UpSet plots (UpSetR [51]) were used to visualize shared and unique core taxa.

Functional profiles were inferred from 16S rRNA sequences with PICRUSt2 [52], generating KEGG Orthology and MetaCyc predictions. Fungal functional guilds were assigned with FUNGuild [31]. Differential enrichment of bacterial functions and fungal guilds across compartments was tested using DESeq2 [53] with adjusted *p* < 0.05.

To compare functional and taxonomic structure, pairwise Aitchison distance matrices were computed for both community composition and PICRUSt2-inferred functions. Mantel and Procrustes (protest) tests were used to assess concordance. A partial Mantel test controlled for shared taxonomic structure. Residual Procrustes analysis visualized environment-driven functional variation after removing taxonomic effects. All tests used 9999 permutations (*p* < 0.05).

Microbial co-occurrence networks were constructed for each compartment using Microeco. Networks were built using SparCC correlations (*p* < 0.05, cutoff = 0.5, abundance filter = 0.0005). Louvain clustering identified modules, and igraph [54] quantified network properties (node count, modularity, positive/negative edges, degree, closeness, betweenness, eigenvector centrality). Taxa roles (network hubs, module hubs, connectors) were identified and visualized with ggraph [55].

Permutation-based tests (coin package) with 9999 resamplings compared node-level centrality metrics among compartments (rhizosphere, root endosphere, leaf endosphere). Geographic distance effects were evaluated using Mantel tests. Community dissimilarities were computed using Euclidean distances, and geographic distances (km) were derived from sampling coordinates using the Haversine formula (geosphere [56]). Mantel correlations used Pearson’s method with 9999 permutations (*p* < 0.05).

## 5. Conclusions

Our results provide one of the most integrative assessments of the *D. antarctica* holobiont to date, revealing how multi-kingdom microbial assemblages are structured across plant compartments and environmentally contrasting islands. We show that compartmentalization remains the dominant force shaping bacterial and fungal communities, while edaphic gradients and geographic context impose additional filtering, particularly in the rhizosphere. Predicted microbial functional potentials further reinforce this hierarchical structure, pointing to a metabolically rich and dynamic rhizosphere, a selective and functionally versatile root endosphere, and a stress-filtered leaf endosphere.

Importantly, these patterns emerge despite the largely clonal genetic background of *D. antarctica* [35], indicating that environmental forces (rather than host genomic variability) play a predominant role in structuring its holobiont. This highlights a system where microbiome flexibility compensates for limited host genetic diversity, supporting plant resilience under extreme Antarctic conditions.

Collectively, our findings advance current understanding of plant–microbe interactions in polar ecosystems by providing a hierarchical conceptual model in which compartmentalization, soil chemistry, geography, microbial stress-adaptive traits, and host genetic uniformity jointly shape holobiont assembly. This framework not only deepens ecological insight into one of Earth’s most climatically constrained terrestrial environments but also offers biological principles relevant for predicting plant–microbe responses to ongoing environmental change.

## Figures and Tables

**Figure 1 plants-14-03657-f001:**
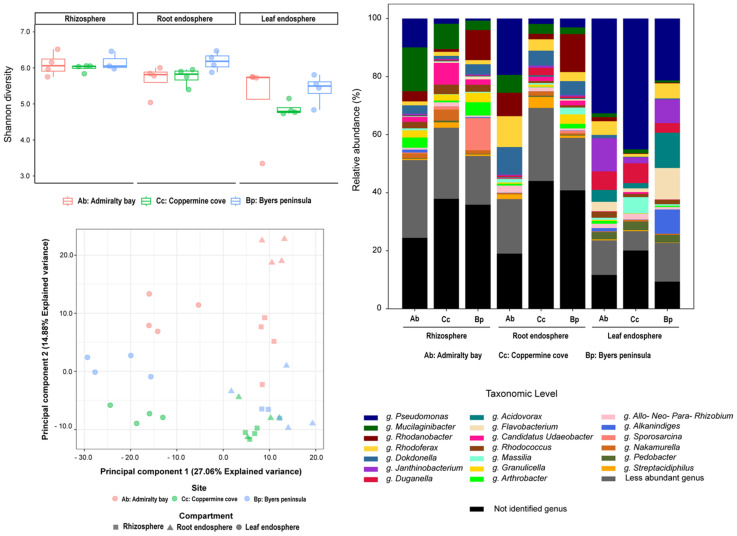
Taxonomic composition, alpha diversity, and beta diversity of bacterial communities associated with *D. antarctica* across three plant compartments (rhizosphere, root endosphere, and leaf endosphere) and three sampling sites in the South Shetland Islands: Admiralty Bay (Ab), Coppermine Cove (Cc), and Byers Peninsula (Bp). **Top Left**: Alpha diversity based on the Shannon index across compartments and locations. **Bottom Left**: Principal Component Analysis (PCA) of bacterial community structure based on 16S rRNA gene sequencing using Aitchison distances derived from centered log-ratio (CLR) transformed data. Shapes represent plant compartments, and colors denote sampling sites. The axes represent the first two components, explaining 27.06% and 14.88% of the total variance, respectively. **Right**: Relative abundance of bacterial genera. Only the most abundant genera are displayed individually; less abundant genera are grouped, and unidentified taxa are shown in black.

**Figure 2 plants-14-03657-f002:**
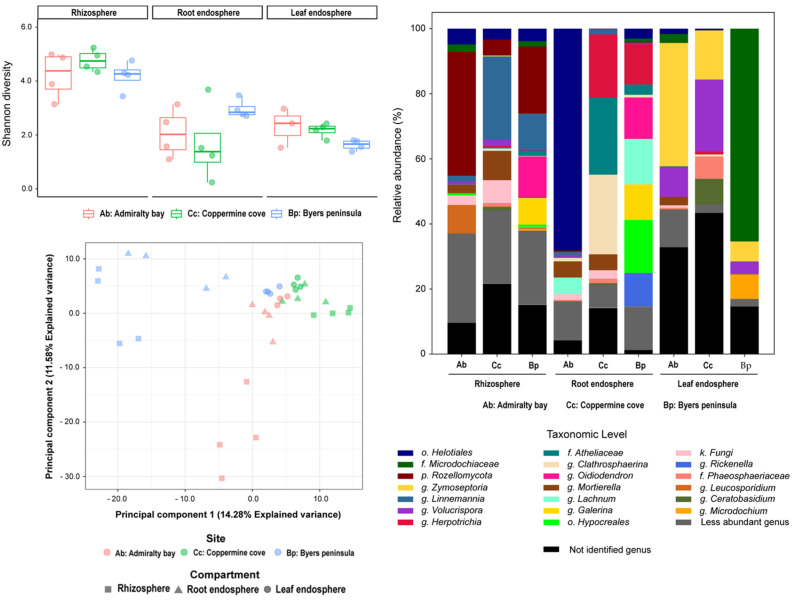
Taxonomic composition, alpha diversity, and beta diversity of fungal communities associated with *D. antarctica* across three plant compartments (rhizosphere, root endosphere, and leaf endosphere) and three sampling sites in the South Shetland Islands: Admiralty Bay (Ab), Coppermine Cove (Cc), and Byers Peninsula (Bp). **Top Left**: Alpha diversity based on the Shannon index across compartments and locations. **Bottom Left**: Principal Component Analysis (PCA) of fungal community structure based on ITS region sequencing using Aitchison distances derived from centered log-ratio (CLR) transformed data. Shapes represent plant compartments and colors denote sampling sites. The first two principal components explained 14.28% and 11.58% of the total variance, respectively. **Right**: Relative abundance of fungal genera. Only the most abundant genera are displayed individually; less abundant genera are grouped, and unidentified taxa are shown in black.

**Figure 3 plants-14-03657-f003:**
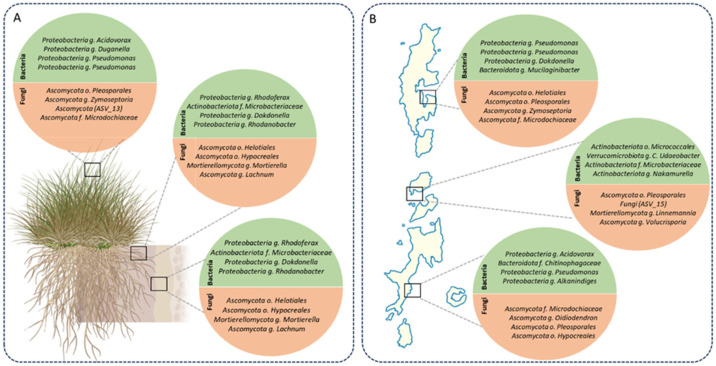
Core microbiomes across compartments and islands in *D. antarctica*. (**A**) Core bacterial (green) and fungal (orange) taxa identified across plant compartments (rhizosphere, root endosphere, and leaf endosphere). (**B**) Core bacterial and fungal taxa consistently detected across the South Shetland Islands (King George, Robert, and Livingston). Circles represent taxonomic groups consistently identified as core members, with representative genera and families indicated. Bacterial communities showed stable cores across compartments and islands, whereas fungal assemblages exhibited stronger compartmental and spatial heterogeneity.

**Figure 4 plants-14-03657-f004:**
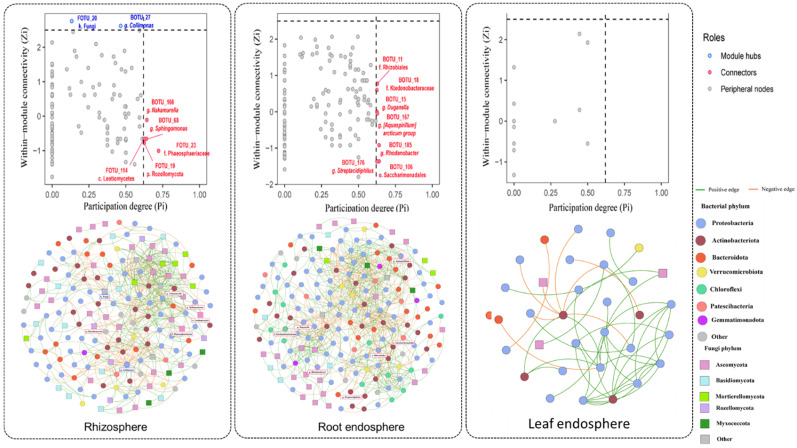
Microbial co-occurrence networks and topological roles across plant compartments of *D. antarctica*. (**Top**) Z–P plots showing the within-module connectivity (Zi) versus the participation coefficient (Pi) for microbial taxa in the rhizosphere, root endosphere, and leaf endosphere. Each dot represents an operational taxonomic unit (ZOTU), with taxa labeled according to their inferred keystone role: module hubs (high Zi), connectors (high Pi), or peripheral nodes. The highlighted taxa included *Lotio-mycetes*, *Rozellomycota*, *Nakamurella*, *Sphingomonas*, *Rhizobiales* and *Burkholderiaceae*. (**Bottom**) Co-occurrence networks were constructed using SparCC correlations (*p* < 0.05, correlation cutoff = 0.5) after applying a zOTU abundance filter of 0.0005. Nodes represent microbial taxa (circles = bacteria, squares = fungi), colored by phylum-level affiliation. Edges indicate positive (green) and negative (red) correlations. The node size is proportional to the degree (number of connections). Networks illustrated marked differences in complexity: the rhizosphere and root endosphere were densely connected with inter-kingdom associations, whereas the leaf endosphere showed sparse, simplified connectivity.

**Figure 5 plants-14-03657-f005:**
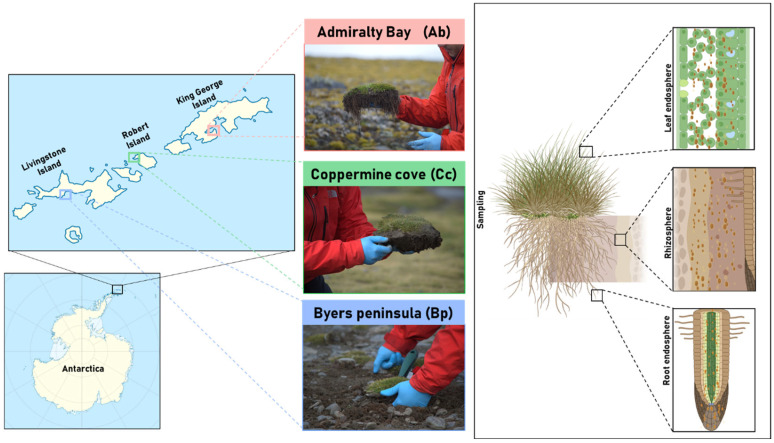
Geographic location and sampling strategy for D. antarctica. Left: Map of the South Shetland Islands showing the three sampling sites: (1) Admiralty Bay (King George Island), (2) Coppermine Cove (Robert Island), and (3) Byers Peninsula (Livingston Island). Right: Schematic representation of the three plant compartments sampled for microbial community analysis: rhizosphere (soil closely adhering to the roots), root endosphere (internal root tissue), and leaf endosphere (internal foliar tissue).

**Table 1 plants-14-03657-t001:** Soil physicochemical analysis across study sites. Values are presented as mean ± standard deviation of duplicate analytical measurements (*n* = 2). Because of the limited replication, data are shown descriptively without formal inferential statistics.

Parameters	Coppermine Cove	Admiralty Bay	Byers Peninsula
Al (cmol^+^ kg^−1^)	5.83 ± 0.06	4.47 ± 0.18	3.85 ± 0.07
Ca (cmol^+^ kg^−1^)	5.22 ± 0.1	0.32 ± 0.06	0.68 ± 0.01
Cu (mg kg^−1^)	1.79 ± 0.01	2.37 ± 0.11	1.23 ± 0.08
Fe (mg kg^−1^)	48.88 ± 0.67	111.89 ± 5.16	239.0 ± 9.9
K (cmol^+^ kg^−1^)	1.17 ± 0.03	0.91 ± 0.06	0.64 ± 0.05
MO (%)	2.0 ± 0.0	2.0 ± 0.0	2.0 ± 0.0
Mg (cmol^+^ kg^−1^)	6.66 ± 0.68	0.17 ± 0.07	0.22 ± 0.01
Mn (mg kg^−1^)	3.13 ± 0.01	0.85 ± 0.01	1.63 ± 0.07
Na (cmol^+^ kg^−1^)	1.88 ± 0.11	0.57 ± 0.06	0.66 ± 0.01
P (mg kg^−1^)	132.0 ± 11.31	315.5 ± 10.61	117.8 ± 0.28
Zn (mg kg^−1^)	7.92 ± 0.06	0.06 ± 0.01	0.16 ± 0.01
pH	5.5 ± 0.01	4.34 ± 0.02	4.34 ± 0.01

## Data Availability

Sequencing data are available in BioProject PRJNA1321289.

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
