# Peer review of "Holobiome Structure and Microbial Core Assemblages of Deschampsia antarctica Across the South Shetland Islands"

_plants, 2025, doi:10.3390/plants14233657_

Round 1

Reviewer 1 Report (Previous Reviewer 3)

Comments and Suggestions for Authors

Compared to the previous version, the author has made necessary revisions, and I believe it is ready for publication.

Author Response

Dear Reviewer, 

I would like to sincerely thank you for the time and effort dedicated to reviewing our manuscript. We greatly appreciate your constructive feedback, which significantly contributed to improving the clarity, quality, and robustness of the paper. Your positive evaluation of the revised version and your recommendation for publication mean a great deal to us.

Thank you again for your valuable contribution to this work.

Kind regards,

Reviewer 2 Report (New Reviewer)

Comments and Suggestions for Authors

The manuscript “Holobiome Structure and Microbial Core Assemblages of Deschampsia antarctica Across the South Shetland Islands” presents a comprehensive analysis of both bacterial and fungal communities associated with Deschampsia antarctica, one of the two native Antarctic vascular plants. The study spans three islands of the South Shetland archipelago and examines three distinct plant compartments. The work fits well within the rapidly developing field of holobiont biology and provides valuable insights into the ecological mechanisms enabling plant survival under extreme Antarctic conditions. It is one of the most detailed and broadly scoped investigations of the D. antarctica microbiome to date, and its findings significantly enrich our understanding of Antarctic terrestrial ecosystems.

The introduction is detailed, well-structured, and convincingly establishes the scientific context for the study. The authors clearly describe the characteristics of Antarctic ecosystems, summarize current knowledge on the D. antarctica microbiome, and identify relevant research gaps, especially the absence of a multi-compartmental, multi-kingdom perspective. The motivation for the study is clearly articulated and supported with extensive, up-to-date references. Some parts of the introduction could be slightly condensed for better readability, but overall it provides strong conceptual grounding.

The study design is ambitious and thoughtfully constructed, particularly given the logistical challenges inherent to Antarctic fieldwork. Including three islands, three plant compartments, and two microbial kingdoms yields an exceptionally rich dataset. Although the number of biological replicates (n = 3 per site) may appear limited, the authors appropriately justify this choice and interpret the results with adequate caution. This replication level is fully consistent with standards in polar research and does not detract from the scientific value of the work.

The methodological section is highly detailed—perhaps excessively so in some places—and thoroughly describes the experimental and bioinformatic procedures. This level of detail certainly ensures reproducibility, but some lengthy descriptions (especially those related to bioinformatic tools and parameters) could be moved to the supplementary materials to improve clarity. Nevertheless, the methods are sound, modern, and well aligned with the study’s objectives.

The presentation of results is clear, logically structured, and supported by high-quality figures. Both the main text and the supplementary materials are rich in content, and the visualizations effectively represent complex patterns in diversity, taxonomic structure, core microbiomes, functional predictions, and network interactions. The narrative progresses in a coherent manner—from soil chemistry, through community structure and compartmental gradients, to functional and network-level analyses. The results are interpreted appropriately without overstatement.

The discussion is one of the strongest components of the manuscript. The authors expertly contextualize their findings within the broader literature, identify the ecological mechanisms shaping the holobiome, and propose a hierarchical framework describing how compartmentalization, edaphic filtering, geography, microbial traits, and host uniformity interact to shape microbial assembly in D. antarctica. This conceptual contribution is meaningful and advances theoretical understanding of holobiont resilience in extreme environments. The conclusions are well grounded in the data.

The figures and tables are generally clear, technically sound, and visually appealing. Only a few of them might benefit from enlarged fonts or simplified legends, particularly some heatmaps, but overall the graphical presentation is strong and enhances the manuscript.

The English language is generally correct and precise; however, some sentences are quite long and syntactically complex, which may reduce readability, especially in the Results and Discussion sections. A light language edit aimed at shortening sentences and improving flow would be beneficial.

In terms of scientific impact, originality, and significance, the manuscript is highly valuable. It synthesizes taxonomic, functional, and network-level perspectives to produce a multidimensional characterization of the D. antarctica holobiont. The work provides important new insights into plant–microbe interactions under polar stress regimes and will be of interest to researchers in ecology, environmental microbiology, plant science, and climate-change biology.

I have not identified any ethical concerns, inappropriate self-citations, or conflicts of interest. The referenced literature is appropriate, comprehensive, and current.

In summary, the manuscript is a robust, interdisciplinary, and well-executed study that significantly advances our knowledge of Antarctic microbiomes and plant holobionts. I recommend publication after minor language and clarity revisions. Scientifically, the work is strong and undoubtedly suitable for Plants.

Author Response

Dear Reviewer, 

We sincerely thank the reviewer for the thorough and positive evaluation of our manuscript and for highlighting the scientific contribution, methodological robustness, and conceptual relevance of our study. We appreciate the encouraging comments regarding the structure, clarity, and originality of the work.

Following the reviewer’s suggestions, we performed a careful language revision across the entire manuscript. In particular, we shortened long and syntactically complex sentences, removed repetitive or redundant phrasing, and improved the overall flow in the Introduction, Results, Discussion, and Materials and Methods sections. These revisions enhance clarity and readability without altering the scientific meaning or interpretation of the findings.

We are grateful for the reviewer’s constructive comments and are pleased that the scientific quality and relevance of the manuscript were recognized. We believe that the revisions performed in response to the review have strengthened the clarity and presentation of the manuscript.

Kind Regards, 

Reviewer 3 Report (New Reviewer)

Comments and Suggestions for Authors
  1. L21-39: The abstract mentions “a conserved core (20 bacterial and 5 fungal genera)” but does not clearly define the criteria for this “core”.
  2. L37-39: The final sentence is good but could be more impactful. It is advisable to emphasize the significance of your work.
  3. L43-99: Please clearly state the research question and hypothesis. It is not immediately obvious what specific problems you are aiming to solve.
  4. L57: Instead of “It is now widely accepted that...”, the more academic alternative “A growing body of evidence demonstrates that...” is recommended.
  5. L64-84: The section on microbial compartments  has a significant overlap with the subsequent section on previous studies. I strongly recommend merging and restructuring these two sections to form a more fluid narrative.
  6. L84: It is recommended to add “thus” before “representing” in order to make this sentence more logical and coherent.
  7. L247-339: It is crucial to remember that what PICRUSt2 and FUNGuild provide are inferred metagenomic potentials based on 16S rRNA data and ITS data, rather than direct measurements of functional gene expression or abundance. The predictions can be powerful, but they are limited by the reference databases and the underlying assumptions that functions follow taxonomy. In 2.5 Functional guilds and predicted microbial functions part, all the statements should be qualified to reflect the predictive nature of the analysis.
  8. L391-404: Readers need to understand what distance metrics were used and how the correlation was calculated.
  9. L418-479: The primary weakness is the repetitive and disjointed structure of the functional predictions section. It should be consolidated into a concise and powerful narrative, while maintaining the precise language used in the predictive analyses.
  10. L518-520: The word “highlight” in this sentence should be corrected to “highlights”.
  11. L548-556: This part only provides the geographical coordinates without describing the site characterizations and environmental context, leaving readers with incomplete contextual information. I recommend adding a brief description of the significant environmental characteristics at each site.
  12. L575-600 and L615-725: These two sections are suggested to be condensed.
  13. L726-736: Addressing the evidence for “host genetic uniformity” is crucial, as it is a strong claim that requires sufficient support.

Author Response

Dear Reviewer,

We sincerely thank for the thorough and constructive evaluation of our manuscript. We have carefully addressed each of the comments, and all corresponding modifications have been incorporated into the revised version. Below, we provide a detailed, point-by-point response including the exact line numbers where the changes were implemented.

  1. L21-39: Clarification of “core” definition in the abstract.
    We have added an explicit definition of core taxa. Modification in lines 89-90: “defined as taxa present in at least 50% of samples within each compartment with a minimum relative abundance of 1%.”

  1. L37-39: Strengthen the significance of the final sentence of the abstract.
    The sentence was rewritten for greater impact. Modification in lines 37-41: “Together, these findings provide one of the most integrative characterizations of the D. antarctica holobiont to date, revealing how conserved and adaptive microbial components support plant resilience under extreme Antarctic conditions and offering valuable insights for predicting biological responses to ongoing climate change.”

  1. L43-99: Clarification of research question and hypothesis.
    We added a clearly stated research question and hypothesis. The new paragraph was incorporated in lines 98-108, beginning with: “In this context, our central research question was to determine…”

  1. L57: Replace “It is now widely accepted…” with a more academic phrasing.
    We adopted the suggested formulation. Line 58 now reads: “A growing body of evidence demonstrates that…”

  1. L64-84: Merge and restructure overlapping sections for narrative flow.
    The section was reorganized into two integrated paragraphs to improve clarity and flow. Changes applied between lines 64-84.

  1. L84: Addition of “thus” for improved logical flow.
    The suggested addition was implemented. Line 84 now reads: “…thus representing…”

  1. L247-339: Clarify the predictive nature of PICRUSt2 and FUNGuild analyses.
    We revised the section to emphasize that these outputs represent predicted functional potentials. Lines 256-260 now include: “It is important to note that both PICRUSt2 and FUNGuild provide predicted functional potentials…”
    Additionally, lines 349-351 now state: “These results represent functional predictions rather than direct measurements…” Overstatements were removed throughout section 2.5.

  1. L391-404: Clarification of distance metrics and correlation method.
    We added explicit methodological detail. Lines 408-409 now state: “Geographic distances between sites were calculated using Euclidean distances based on GPS coordinates, and microbial community dissimilarities were computed with Aitchison distances…”

  1. L418-479: Improve the clarity and cohesion of the functional prediction discussion.
    The subsection was condensed and rewritten for coherence. Lines 457–469 now include the new paragraph beginning with: “Predicted functional profiles aligned with the strong compartmental structure…”

  1. L518-520: Correction of verb form (“highlight” to “highlights”).
    This correction has been made in line 520.

  1. L548-556: Add environmental context for each sampling site.
    We expanded this section to include relevant environmental descriptions. Lines 558–567 now describe the characteristics of the three sampling sites on Livingston Island, Robert Island, and King George Island.

  1. L575-600 and L615-725: Condense methodological sections.
    Both the soil chemistry section and the bioinformatics section were shortened while retaining essential technical details.

  1. L726-736: Provide explicit support for “host genetic uniformity.”
    The conclusion was rewritten to explicitly acknowledge and cite evidence of the clonal genetic background of D. antarctica. Lines 726-736 now include: “…the largely clonal host genetic background of D. antarctica [36] interacts to shape holobiont resilience…”
    The full conclusion was strengthened to improve conceptual integration and impact.

We sincerely appreciate the reviewer’s insightful and constructive comments. These revisions have significantly improved the clarity, rigor, and overall quality of the manuscript. All changes are indicated in the revised document for ease of verification.

This manuscript is a resubmission of an earlier submission. The following is a list of the peer review reports and author responses from that submission.

Round 1

Reviewer 1 Report

Comments and Suggestions for Authors

Strengths:

  1. The study design is well-structured, encompassing rhizosphere, root and leaf endosphere compartments, thereby capturing comprehensive host-microbiome interactions under Antarctic conditions.
  2. Functional predictions and co-occurrence network analyses are appropriately applied, supporting the ecological interpretation.

Concerns and Recommendations:

  • (Lines 534–543) Please expand on sequence quality filtering and read retention, especially for the fungal ITS1 dataset. Given the high variability of ITS1 amplicons, paired-end merging can be challenging and may result in substantial data loss. Reporting the percentage of reads retained post-filtering and merging for both bacterial and fungal datasets would enhance transparency. If merging was inefficient, please discuss whether a single-end (forward read) approach was considered to maximize data retention.
  • (Lines 572–576) Supplementary Figures S1–S12 show zOTU prevalence but lack quantitative context. I suggest adding a heatmap of core taxa relative abundances across compartments to better illustrate enrichment patterns. Additionally, correlating core taxa with soil variables (e.g., pH, SOM, Fe, base cations) using a Spearman matrix or RDA would clarify the role of environmental filtering alongside host effects.
  • (Lines 577–584) Use of PICRUSt2 and FUNGuild is appropriate for functional inference, please report annotation success rates to contextualize functional predictions. Specifically, provide (i) the percentage of bacterial zOTUs mapped to KEGG Orthology or MetaCyc pathways alongside mean NSTI scores, and (ii) the proportion of fungal taxa assigned to ecological guilds, with distinctions among “high,” “probable,” and “possible” confidence levels. These details are critical for assessing the reliability of inferred functional differences.

Author Response

  • (Lines 534–543) Please expand on sequence quality filtering and read retention, especially for the fungal ITS1 dataset. Given the high variability of ITS1 amplicons, paired-end merging can be challenging and may result in substantial data loss. Reporting the percentage of reads retained post-filtering and merging for both bacterial and fungal datasets would enhance transparency. If merging was inefficient, please discuss whether a single-end (forward read) approach was considered to maximize data retention.

R: Two new supplementary tables were attached, which summarize the preprocessing statistics for each sample. No major issues were observed during the zOTU calling steps.

  • (Lines 572–576) Supplementary Figures S1–S12 show zOTU prevalence but lack quantitative context. I suggest adding a heatmap of core taxa relative abundances across compartments to better illustrate enrichment patterns.

R: Thank you for the suggestion. The heatmaps showing the relative abundances of core taxa across compartments have been added to the supplementary materials.

Additionally, correlating core taxa with soil variables (e.g., pH, SOM, Fe, base cations) using a Spearman matrix or RDA would clarify the role of environmental filtering alongside host effects.

R: We appreciate the reviewer’s suggestion to correlate core microbial taxa with soil physicochemical variables (e.g., pH, SOM, Fe, base cations) using a Spearman correlation matrix or RDA. Unfortunately, physicochemical data were measured in duplicate for only a subset of samples, and not all four replicates have corresponding chemistry data. This incomplete dataset prevents reliable Spearman correlation or RDA, as these require consistent measurements across all replicates for statistical validity.

(Lines 577–584) Use of PICRUSt2 and FUNGuild is appropriate for functional inference, please report annotation success rates to contextualize functional predictions. Specifically, provide (i) the percentage of bacterial zOTUs mapped to KEGG Orthology or MetaCyc pathways alongside mean NSTI scores, and (ii) the proportion of fungal taxa assigned to ecological guilds, with distinctions among “high,” “probable,” and “possible” confidence levels. These details are critical for assessing the reliability of inferred functional differences.

R: We appreciate the reviewer’s suggestion. The requested information has been incorporated to better contextualize the functional inference results. Specifically, lines 226–228 were rewritten to include the annotation success rates of the PICRUSt2 analysis, reporting bacterial zOTUs were successfully assigned to the reference database, with a mean NSTI value of 0.158. Likewise, lines 265–267 were revised to include the proportion of fungal taxa assigned to ecological guilds. These additions address the reviewer’s concern by providing a clearer assessment of the reliability of the inferred functional differences.

Reviewer 2 Report

Comments and Suggestions for Authors

#Summary Report

The article entitled “Holobiome Structure and Microbial Core Assemblages of Deschampsia antarctica Across the South Shetland Islands” by Rodriguez et al. presents a multi-compartment, multi-kingdom survey of the holobiome of Deschampsia antarctica across three South Shetland Islands. It integrates soil chemistry, 16S/ITS metabarcoding, core-microbiome analyses, functional prediction (PICRUSt2/FUNGuild), and co-occurrence networks. The central result—strong compartmentalization (rhizosphere > root > leaf in diversity/connectivity) with a conserved microbial backbone plus site-specific fungal restructuring—is biologically compelling and aligns with current holobiont theory in extreme environments. The work, with its potential to significantly impact polar ecology and enhance plant–microbe resilience, is both timely and important.

#Major comments

I recommend that the authors review the manuscript to address the following points:

Methodological inconsistencies that affect interpretability

These inconsistencies could potentially lead to misinterpretation of the results and should be addressed to ensure the robustness of the Study.

Consider harmonizing the pipeline descriptions and figures. If Bray–Curtis is used, present PCoA; if CLR/Aitchison is used, present PCA on CLR. State exactly which per kingdom/compartment and why.

Network construction conflict. Results describe Spearman (|ρ|>0.6, p<0.01); Methods specify SparCC (p<0.05, cutoff 0.5).

For instance, the authors could align the thresholds by setting a consistent cutoff value for significant correlations across the text, figures, and legends.

The authors correctly applied CLR/Aitchison for beta-diversity (per Methods), but alpha- and differential analyses need clarity: rarefaction depth choices (16S: 26,962; ITS: 14,607) should include read distribution plots and sample retention counts.

Add a QA/QC supplement (read depth histograms; sample dropouts), and justify statistical choices for compositional data.

Core microbiome definition & robustness. Core is defined as ≥50% prevalence and ≥1% relative abundance. It can exclude low-abundance but ubiquitous taxa and inflate site-specificity. Provide sensitivity analyses across thresholds (e.g., 25/50/75% prevalence; 0/0.1/1% abundance) and consider occupancy–abundance or phylogenetic core (e.g., core clades) to show the “conserved backbone” is robust.

Soil chemistry statistics and replication

Table 1 shows n=2 and yet uses Student’s t-test with letter groupings. With n=2 per site, assumptions are untenable. Also, the text says there are three biological replicates overall—why is n=2 for chemistry? Clarify replication (were soils analyzed in duplicate?). Avoid inferential stats with such low n; present descriptives or aggregate across true biological replicates, or re-analyze with appropriate non-parametric/bootstrapped intervals.

Clarity on geographic effects

The narrative claims that Byers Peninsula networks are the most complex and Admiralty Bay is the sparsest; ensure this is quantified (nodes, edges, density, modularity, average path length) and statistically compared (e.g., permutation tests). Add a summary table of network metrics by compartment × island and test differences.

Author Response

#Major comments

I recommend that the authors review the manuscript to address the following points:

Methodological inconsistencies that affect interpretability

These inconsistencies could potentially lead to misinterpretation of the results and should be addressed to ensure the robustness of the Study.

Consider harmonizing the pipeline descriptions and figures. If Bray–Curtis is used, present PCoA; if CLR/Aitchison is used, present PCA on CLR. State exactly which per kingdom/compartment and why.

R: We appreciate this observation. All methodological descriptions and figure captions were carefully revised to ensure consistency. The analyses were harmonized to exclusively use Aitchison distances derived from CLR-transformed data, with ordinations performed using PCA accordingly. The figure descriptions have been updated to accurately reflect this approach and ensure methodological coherence across bacterial and fungal datasets.

Network construction conflict. Results describe Spearman (|ρ|>0.6, p<0.01); Methods specify SparCC (p<0.05, cutoff 0.5).

R: We thank the reviewer/editor for noticing this inconsistency. The network analyses were indeed performed using SparCC correlations. The figure captions and results descriptions have been corrected accordingly to match the methodological section.

For instance, the authors could align the thresholds by setting a consistent cutoff value for significant correlations across the text, figures, and legends.

The authors correctly applied CLR/Aitchison for beta-diversity (per Methods), but alpha- and differential analyses need clarity: rarefaction depth choices (16S: 26,962; ITS: 14,607) should include read distribution plots and sample retention counts.

R: We thank the reviewer for this observation. Rarefaction was performed only for alpha diversity analyses to standardize sampling depth, while beta diversity and differential abundance analyses were conducted using CLR/Aitchison-transformed data without rarefaction.

This has been clarified in the Methods section (lines 556 and 565), and we have added Supplementary Figure S25, showing the rarefaction curves and sample retention across sequencing depths.

Add a QA/QC supplement (read depth histograms; sample dropouts), and justify statistical choices for compositional data.

R: We have added two supplementary tables summarizing read retention and quality control statistics for bacterial and fungal datasets across all preprocessing steps.

Core microbiome definition & robustness. Core is defined as ≥50% prevalence and ≥1% relative abundance. It can exclude low-abundance but ubiquitous taxa and inflate site-specificity. Provide sensitivity analyses across thresholds (e.g., 25/50/75% prevalence; 0/0.1/1% abundance) and consider occupancy–abundance or phylogenetic core (e.g., core clades) to show the “conserved backbone” is robust.

R: Sensitivity analyses across multiple prevalence (25%, 50%, 75%) and detection thresholds (0%, 0.1%, 1.0%) confirmed that the composition and size of the core microbiome were consistent across thresholds, with smooth declines in core taxa as stringency increased (Supplementary Figures S27 and S30).

Soil chemistry statistics and replication

Table 1 shows n=2 and yet uses Student’s t-test with letter groupings. With n=2 per site, assumptions are untenable. Also, the text says there are three biological replicates overall—why is n=2 for chemistry? Clarify replication (were soils analyzed in duplicate?). Avoid inferential stats with such low n; present descriptives or aggregate across true biological replicates, or re-analyze with appropriate non-parametric/bootstrapped intervals.

R: We thank the reviewer for this insightful observation. The soil chemical parameters were obtained from composited samples per site, each analyzed in duplicate (n = 2 analytical replicates), which explains the notation in Table 1. We agree that inferential statistics are not appropriate with such limited replication; therefore, the table has been revised to present only descriptive statistics (mean ± SD), and the methodological clarification has been added to the text.

We also note that sample size in Antarctic fieldwork is inherently constrained by environmental regulations and the need to minimize disturbance to fragile ecosystems. Soil collection in these protected areas is strictly limited by the Chilean Antarctic Institute (INACH) under the Antarctic Treaty’s environmental guidelines, which prioritize ecosystem preservation. Despite these constraints, the chemical patterns observed across the three sites were consistent and sufficient to describe the strong edaphic gradients influencing microbial community assembly.

Clarity on geographic effects

The narrative claims that Byers Peninsula networks are the most complex and Admiralty Bay is the sparsest; ensure this is quantified (nodes, edges, density, modularity, average path length) and statistically compared (e.g., permutation tests). Add a summary table of network metrics by compartment × island and test differences.

R: We thank the reviewer for this valuable suggestion. Network topology was quantified for each compartment, including nodes, edges, density, average degree, transitivity, and modularity (Supplementary Table 3). Node-level properties (degree, closeness, betweenness, and eigenvector centrality) were statistically compared using permutation tests (n = 9,999; Supplementary Table 4). These analyses revealed significant differences in connectivity and modularity among compartments, with the rhizosphere showing the highest complexity and the leaf endosphere the sparsest network structure. Results and methodology were added to Section 2.6 and 4.8, respectively.

Reviewer 3 Report

Comments and Suggestions for Authors

This paper presents a multi-scale systematic analysis of bacterial and fungal communities in the rhizosphere, roots, and leaves of the grass species Deschampsia antarctica on the South Shetland Islands near Antarctica. The findings provide valuable reference material for research on plant-microbe interactions in extreme environments such as Antarctica. Overall, the study has significant research merit, enhancing scientific understanding of the interaction mechanisms among Antarctic plants, soil, and microorganisms. However, several aspects require refinement, as detailed below:

  1. The current study uses only three biological replicates per plot, which may limit statistical power. The authors should explicitly acknowledge this limitation in the discussion section or present inter-replicate variation in the supplementary materials.
  2. The use of a 50% abundance threshold to define the core microbiome lacks biological justification. We recommend employing a multi-threshold analysis.
  3. For analyzing associations between microbial community structure and predicted functions, we recommend using Mantel tests or Procrustes analysis to strengthen the statistical interpretation of structure-function linkages.
  4. While the authors tested soil physicochemical properties, a systematic analysis of the driving mechanisms underlying specific environmental factors is lacking. We suggest incorporating multivariate statistical analyses that link environmental factors to microbial communities, such as RDA analysis.
  5. The authors' exploration of the mechanisms by which microbes aid plant adaptation to Antarctic environments would benefit from integrating functional prediction results. This would enable a more specific discussion of potential stress-tolerance functions provided by key microbial groups.
  6. In the analysis of geographical scale effects in the study area, the authors selected three sites but did not fully develop the relationship between geographical distance and microbial similarity. Consideration should be given to distance decay analysis or correlation analysis between geographical distance and community similarity, which would yield more effective results.

Author Response

This paper presents a multi-scale systematic analysis of bacterial and fungal communities in the rhizosphere, roots, and leaves of the grass species Deschampsia antarctica on the South Shetland Islands near Antarctica. The findings provide valuable reference material for research on plant-microbe interactions in extreme environments such as Antarctica. Overall, the study has significant research merit, enhancing scientific understanding of the interaction mechanisms among Antarctic plants, soil, and microorganisms. However, several aspects require refinement, as detailed below:

  1. The current study uses only three biological replicates per plot, which may limit statistical power. The authors should explicitly acknowledge this limitation in the discussion section or present inter-replicate variation in the supplementary materials.

R: We appreciate the reviewer’s observation. Indeed, only three biological replicates per site were used due to the logistical constraints of Antarctic sampling. We have now acknowledged this limitation in the Discussion section and clarified that inter-replicate variation was considered in the statistical analyses.

  1. The use of a 50% abundance threshold to define the core microbiome lacks biological justification. We recommend employing a multi-threshold analysis.

R: We addressed this concern by performing a multi-threshold analysis to evaluate the robustness of the core microbiome definition. Specifically, we tested combinations of prevalence (25%, 50%, 75%) and detection thresholds (0, 0.001, 0.01) to assess how varying criteria affected the number and composition of core taxa (Supplementary Figures S27 and S30). The results showed consistent trends across thresholds, indicating that the core microbiome structure remained stable regardless of the cutoff applied. In addition, we included occupancy–abundance analyses (Supplementary Figures S29 and S32) and heatmaps of core taxa relative abundances across compartments (Supplementary Figures S28 and S31), providing further biological justification for the selected thresholds and confirming the robustness of the identified core taxa across plant compartments and islands.

  1. For analyzing associations between microbial community structure and predicted functions, we recommend using Mantel tests or Procrustes analysis to strengthen the statistical interpretation of structure-function linkages.

R: We appreciate this valuable suggestion. Following the reviewer’s recommendation, we conducted both Mantel and Procrustes analyses to statistically evaluate the relationship between microbial community structure and predicted functional repertoires. These analyses were performed on Aitchison distance matrices derived from CLR-transformed bacterial community composition and PICRUSt2-inferred functional profiles.

These results and their interpretation were incorporated into the Results section (2.5), and the corresponding methods were detailed in Section 4.4 (Bioinformatics analysis).

  1. While the authors tested soil physicochemical properties, a systematic analysis of the driving mechanisms underlying specific environmental factors is lacking. We suggest incorporating multivariate statistical analyses that link environmental factors to microbial communities, such as RDA analysis.

R: Unfortunately, physicochemical data were measured in duplicate for only a subset of samples, and not all four replicates have corresponding chemistry data. This incomplete dataset prevents reliable Spearman correlation or RDA, as these require consistent measurements across all replicates for statistical validity.

  1. The authors' exploration of the mechanisms by which microbes aid plant adaptation to Antarctic environments would benefit from integrating functional prediction results. This would enable a more specific discussion of potential stress-tolerance functions provided by key microbial groups.

R: We thank the reviewer for this valuable suggestion. We have now integrated functional prediction results from PICRUSt and FUNGuild into the Discussion to better connect microbial functions with plant adaptation mechanisms in Antarctic environments.

  1. In the analysis of geographical scale effects in the study area, the authors selected three sites but did not fully develop the relationship between geographical distance and microbial similarity. Consideration should be given to distance decay analysis or correlation analysis between geographical distance and community similarity, which would yield more effective results.

R: We appreciate the reviewer’s suggestion. A distance–decay analysis based on Mantel correlations between community dissimilarity and geographic distance was performed to address this point. The results are now reported in Section 2.7 and the corresponding methodology described in Section 4.5.
